# Osteoporosis among Postmenopausal Women in Jordan: A National Cross-Sectional Study

**DOI:** 10.3390/ijerph19148803

**Published:** 2022-07-20

**Authors:** Rami Saadeh, Duaa Jumaa, Lina Elsalem, Anwar Batieha, Hashem Jaddou, Yousef Khader, Mohammed El-Khateeb, Kamel Ajlouni, Mohammed Z. Allouh

**Affiliations:** 1Department of Public Health and Community Medicine, Faculty of Medicine, Jordan University of Science and Technology, Irbid 22110, Jordan; rasaadeh@just.edu.jo (R.S.); opt.duaa92@gmail.com (D.J.); batieha@just.edu.jo (A.B.); jaddou@just.edu.jo (H.J.); yskhader@just.edu.jo (Y.K.); 2Department of Pharmacology, Faculty of Medicine, Jordan University of Science and Technology, Irbid 22110, Jordan; lmelsalem@just.edu.jo; 3The National Center (Institute) for Diabetes, Endocrinology and Genetics, University of Jordan, Amman 11942, Jordan; mkhateeb@ju.edu.jo (M.E.-K.); ajlouni@ju.edu.jo (K.A.); 4Department of Anatomy, Faculty of Medicine, Jordan University of Science and Technology, Irbid 22110, Jordan; 5Department of Anatomy, College of Medicine and Health Sciences, United Arab Emirates University, Al Ain 15551, United Arab Emirates

**Keywords:** osteoporosis, vitamin D, calcium, menopause

## Abstract

Osteoporosis is considered a widespread health problem that affects senior citizens, particularly older women, after the menopause. This national study aimed to estimate the prevalence of osteoporosis among Jordanian postmenopausal women and to determine the association of demographic and nutritional factors, such as calcium and vitamin D supplement intake, with osteoporosis in postmenopausal women. A cross-sectional study was conducted among 884 postmenopausal women aged ≥50 years. A multistage sampling technique was used to select participants from three geographic regions of Jordan (north, middle, and south). The data were collected from the participants by a team of field researchers comprising men and women through a standard questionnaire. The prevalence of osteoporosis was 19.8% among postmenopausal Jordanian women. The study results showed that age (*p* ˂ 0.001), geographic region (*p* = 0.019), occupation (*p* = 0.002), and educational level (*p* = 0.001) were significantly associated with osteoporosis. Moreover, osteoporosis was significantly associated with calcium and vitamin D supplement intake (*p* < 0.05). There is a high prevalence of osteoporosis among postmenopausal Jordanian women. Therefore, there is a need to educate women at this age, and probably at an earlier age, to prevent or reduce the development of osteoporosis.

## 1. Introduction

Osteoporosis is a bone disorder characterized by a reduction in bone mineral density and the deterioration of bone tissue, resulting in skeletal fragility and fracture susceptibility [1]. Currently, osteoporosis is considered a global health problem that occurs at high rates in western countries, Asia, and Latin America [2]. According to the World Health Organization (WHO), osteoporosis is described as the “silent disease of the century,” with major economic and social impacts [3]. It is well documented that osteoporosis mainly affects older adults, with a higher prevalence among women [4]. In particular, postmenopausal women are at a higher risk than other women, with one-half of all postmenopausal women experiencing osteoporosis [5,6,7,8].

Bone fractures, the main sign of osteoporosis, are a major cause of human suffering and a financial burden on healthcare systems [9]. Approximately 9 million cases of osteoporotic fractures are reported each year, with expectations to increase by four times by 2050 [10]. Based on the International Osteoporosis Foundation’s reports, one in five men and one in three women aged ≥50 years suffer from bone fracture [11,12,13]. Of note, 25% of postmenopausal women with osteoporosis had a vertebral deformity and 15% had a hip fracture [14]. 

The development of osteoporosis in postmenopausal women has been attributed to many risk factors, such as physiological and hormonal changes, including a progressive reduction in estrogen levels that affects calcium absorption [15,16,17]. In turn, a reduction in calcium levels has a negative impact on bone density and bone restructuring. Reduced calcium absorption can lead to the gradual loss of bone mass (osteopenia) and eventually osteoporosis with a higher possibility of fractures [18]. 

Lifestyle factors, including calcium and vitamin D supplementation, physical activity, smoking, and body weight are also described as risk factors for osteoporosis in postmenopausal women [19]. Calcium and vitamin D are well known as the most important elements needed for bone health [20]. Intake deficiencies among postmenopausal women contribute to inefficient bone restructuring [21,22], and increase the risk of bone fractures [23]. The main sources of calcium are dairy products (yogurt, cheese, milk) and green vegetables [24], while dairy products, foods fortified with vitamin D, and sunlight exposure are the main sources of vitamin D [25]. Mounting evidence points to the role of calcium and vitamin D supplementation in reducing bone loss and fractures among older people [26,27,28]. Accordingly, the US National Institute of Health (NIH) recommended that calcium intake for postmenopausal women should be 1500 mg/day [29]. In addition, bone loss prevention strategies in postmenopausal women should include the prescription of vitamin D and calcium supplements [20,30]. 

Many international epidemiologic studies have extensively analyzed and explained potential osteoporosis risk factors, which include demographics, social information, medical history, alcohol consumption, smoking habits, maternal and paternal history of bone fracture after the age of 50 years, calcium intake, and present and past physical activities. However, the conclusions obtained from these studies, which were based on various ethnic groups, are controversial [31,32,33,34,35,36]. 

To the best of our knowledge, limited evidence is available regarding the prevalence and risk factors of osteoporosis in Jordan. Findings from one prospective cross-sectional study reported that 43.3% of postmenopausal women had osteoporosis [37]. Another study revealed that one-third (29.6%) of Jordanian women with a mean age of 53.23 years had osteoporosis [37]. A study with a large sample (*n* = 1079) of Jordanian postmenopausal women aged between 45 and 84 years showed that the prevalence of osteoporosis was 37.5% [5]. Furthermore, osteoporosis was significantly associated with age, physical activity, sun exposure, high parathyroid hormone level, calcium intake, caffeine intake, and marital status [5]. In comparison, another study reported that the prevalence of osteoporosis was approximately 16.7% among 333 postmenopausal women and was significantly associated with low peak bone mass, relatively low vitamin D, high body mass index, and genetic and dietary factors [6]. 

The limitation of data in Jordan regarding osteoporosis among postmenopausal women and its associated lifestyle factors signifies the importance of this national study, which aimed to (1) estimate the prevalence of osteoporosis among Jordanian postmenopausal women and (2) determine the effects of demographic and nutritional factors, such as calcium and vitamin D intake, on osteoporosis in postmenopausal women. 

## 2. Methods

### 2.1. Ethical Considerations

This study was approved by the research ethics committee of the National Center for Diabetes, Endocrinology, and Genetics (Amman, Jordan) and the Institutional Review Board of Jordan University of Science and Technology (IRB # 414-2022). A support letter was obtained from the Jordanian Ministry of Health to guarantee the collaboration of the staff working in health centers where the data were collected. 

### 2.2. Study Design and Sampling

A cross-sectional study was conducted among postmenopausal women aged ≥50 years. The target population was selected from three geographic regions of Jordan (north, middle, and south). A multistage sampling technique was used to enroll a total of 884 postmenopausal women [38]. The sampling was proportional to the size of the target area to cover the entire target population, which included the 12 governorates of Jordan of the three geographic regions of Jordan. At stage one of the sampling, clusters were randomly selected from each city using computer-generated numbers, and on the second stage, households were randomly selected within a selected area using a systematic sampling technique. 

Public health directors in every governorate of Jordan were asked to select one to three health centers that are large with expanded services as representative health centers in each governorate. The study team (25 persons) performed the procedures required for the study [38]. 

### 2.3. Data Collection 

The data used and analyzed in this study were derived from a survey conducted by the National Center for Diabetes, Endocrinology, and Genetics, Jordan Ministry of Health, and Jordan University of Science and Technology. A team of field researchers (comprising men and women) was recruited to collect data from participants, and they went from door to door to invite a systematic sample of households in the catchment area of the selected center. The team explained the study to all participants and responded to all questions from them [38]. 

The team asked the participants who agreed to participate in the study to visit the health center early in the morning for a health examination and interview. The study team worked every day (except Fridays) to obtain the highest possible response rate. All researchers underwent a training workshop that lasted two days prior to data collection. The workshop included explaining the aim and procedures of the study and developing skills to interview the participants [38].

### 2.4. Research Instrument 

Each woman in this study was interviewed using a standard paper-based questionnaire to collect information on demographic data, nutrition variables, and bone problems. The questionnaire was pre-tested and validated and included both dichotomous and multiple-choice questions. The original questionnaire was multi-purpose, containing many questions. However, only those related to the current study were included in the analysis. The survey questions included the following: Demographic information: Region, age, educational level, marital status, and occupation data were collected.Nutrition-related information: The participants were asked whether they were currently or previously taking vitamin D and calcium. The participants were also asked whether they had vitamin D deficiency. Finally, the participants were asked other nutritional questions, such as the total number of meals they had in the week before the study interview and the number of times they consumed vegetables or fruits in the week before the study interview.Osteoporosis information: The participants were asked whether they had been diagnosed with osteoporosis (self-report).

### 2.5. Statistical Analysis 

All analyses were conducted using IBM SPSS software (version 24, IBM, Armonk, NY, USA). The responses were analyzed using descriptive statistics, including mean, frequency, and percentage. The chi-square test for independence was used for categorical variables. Moreover, binary logistic regression models were used to investigate the association between osteoporosis and the demographic and nutritional factors. Statistical significance was set at *p* < 0.05. 

## 3. Results

### 3.1. Demographic Information 

This study included a total of 884 postmenopausal women aged between 50 and 80 years. The demographic characteristics of the participants are presented in Table 1. The age of most women (73.4%) ranged from 50 to 59 years. The percentage of participants living in the middle region of Jordan, representing the capital area, was higher (44.0%) than that of participants living in the northern and southern regions. Most of the participants were married (69.8%), had completed primary school (44.5%), and were unemployed (77.0%) (Table 1).

### 3.2. Osteoporosis among Participants

The prevalence of osteoporosis was 19.8%. Table 2 shows the prevalence of osteoporosis according to demographic variables. As per the region, the prevalence of osteoporosis was 24.5% in the northern region, 19.5% in the middle region, and 14.4% in the southern region. The prevalence of osteoporosis was 14.8% among those aged 50–59 years, 32.6% among those aged 60–69 years, and 38.1% among those aged ≥70 years. Osteoporosis was less prevalent among married women (18.3%) than among unmarried women (23.4%). Unemployed participants had the highest prevalence of osteoporosis (22.2%); the prevalence was low among retired (13.3%) and employed (7.4%) participants. The prevalence of osteoporosis among women was the highest (24.7%) among those who only completed primary school (Table 2). 

Overall, the prevalence of osteoporosis (*p* = 0.083) was not significantly different between the married and unmarried individuals. Contrastingly, the difference in the prevalence of osteoporosis was significant (*p* = 0.001) among participants with different levels of education (primary and secondary school, college degree, or higher). Additionally, age (*p* ˂ 0.001), region (*p* = 0.019), occupation (*p* = 0.002), and educational level (*p* = 0.001) were significantly associated with osteoporosis (Table 2).

### 3.3. Association between Osteoporosis and Demographic Characteristics 

Table 3 shows that osteoporosis was significantly associated with certain demographic characteristics. Participants who lived in the middle region of Jordan were more likely to have osteoporosis (odds ratio (OR) = 1.943, 95% confidence interval (CI) = 1.216–3.103, *p*-value = 0.005) than those who lived in the northern region. Women aged 50–59 years were less likely to have osteoporosis (OR = 0.363, 95% CI = 0.179–0.737, *p* = 0.005) than those aged ≥70 years. The study period and occupation were not significantly associated with osteoporosis (Table 3).

### 3.4. Osteoporosis and Nutrition Variables

Most women (59.0% and 67.5%, respectively) in the study had never used vitamin D or calcium supplements. The percentage of women with vitamin D deficiency was 30.7%. Most participants were consuming 1–10 meals in the week before the interview (92.9%). Almost three-quarters of the participants consumed vegetables or fruits 0–5 times during the week before the study interview (71.5%). 

Table 4 shows the prevalence of osteoporosis according to nutritional variables. The prevalence of osteoporosis was 31.3% among women who were currently taking vitamin D supplements, 22.2% among those who previously took vitamin D supplements, and 15.5% among those who never took vitamin D supplements. The prevalence of osteoporosis was 56.8% among women who were currently taking calcium supplements, 31.2% among women who previously took calcium supplements, and 10.7% among those who never took calcium supplements. Participants who were diagnosed with vitamin D deficiency had the highest prevalence of osteoporosis (26.6%) among the study sample. The prevalence of osteoporosis among women who consumed 1–10 meals in the week before the study interview was 20.1%, which is higher than the prevalence of osteoporosis among women who consumed 11–21 meals. The prevalence of osteoporosis was 19.3% for those who consumed vegetables or fruits 0–5 times during the week before the study interview and 20.7% for those who consumed them 6–11 times during the week before the study interview (Table 4). 

Overall, the difference in the prevalence of osteoporosis was significant in the case of supplementary calcium and vitamin D intake (*p* ˂ 0.05). The results showed a significant association between osteoporosis and vitamin D intake (*p* ˂ 0.001), calcium intake (P˂0.001), and vitamin D deficiency (*p* = 0.001).

### 3.5. Association between Osteoporosis and Nutrition-Related Factors

Table 5 shows the association between osteoporosis and nutrition-related variables. The results show a significant association between osteoporosis and supplementary calcium intake. Furthermore, the results show that vitamin D intake one week before the study interview was significantly associated with osteoporosis. Participants who took calcium supplements previously (OR = 10.272, 95% CI = 5.791–18.219, *p* ˂ 0.001) and those who never took calcium supplements (OR = 3.823, 95% CI = 2.501–5.844, *p* ˂ 0.001) were more likely to have osteoporosis than those currently taking calcium supplements. Participants who previously consumed vitamin D supplements were more likely to have osteoporosis (OR = 2.161, 95% CI = 1.227–3.807, *p* = 0.008) than those who were currently taking vitamin D supplements. Vitamin D deficiency was not significantly associated with osteoporosis (Table 5).

## 4. Discussion

In this national study, the prevalence of osteoporosis among Jordanian postmenopausal women was 19.8%. A cross-sectional study reported that almost one-third (29.6%) of Jordanian women with a mean age of 53.23 years had osteoporosis [37]. Comparatively, regional studies show different prevalence rates; for example, studies in Turkey [39] and Saudi Arabia [40] reported prevalence rates of 16.2% and 63.63%, respectively. A cross-sectional study conducted among 31,769 participants (aged 45–86 years) in China reported that the prevalence of osteoporosis was 23.9% among women [41]. However, there is a disparity in the results among these studies, which might be attributed to differences in the patient selection criteria, sample size, study design, or the diagnostic mechanism used. In addition, patient-related factors, such as genetic background and nutritional and physical activity factors, might contribute to differences in the prevalence of osteoporosis.

Regarding patient demographics, the data of this study showed an association between age and osteoporosis. This is in accordance with the findings of other studies that considered age as a high-risk factor for the development of osteoporosis [42] and bone fractures [43]. We also found a significant association between educational levels and the risk of osteoporosis. The prevalence of osteoporosis among women who completed primary school was higher than that among women with higher levels of education. This supports findings from a previous study in Denmark [43], in which an inverse relationship was found between educational levels and the risk of fracture among people aged 40–59 years. One possible explanation is that women with lower education levels tend to have lesser knowledge about osteoporosis and its related risk factors. Additionally, women with lower education levels usually have lower incomes, which might be reflected in lower food quality and poorer lifestyles. However, a prospective study conducted in the United States [44] and a case-control study from Sweden [45] reported no significant association between educational levels and the risk of fracture. 

A significant association was found between occupation and the risk of osteoporosis. The prevalence of osteoporosis was the highest among unemployed participants. Unemployed women might be less exposed to sunlight than are employed women; as unemployed women stay at home for long periods, they might have lower levels of vitamin D, an essential element for bone health. Furthermore, they might have less physical activity; notably, low physical activity is a risk factor for osteoporosis [17]. A case-control study conducted on the socioeconomic aspects of fractures within universal public healthcare in Denmark concluded that the risk of fracture was significantly reduced among employed participants aged 40–59 years [43]. However, the same study reported no significant association between the risk of fracture and type of employment for those aged ˃60 years, which was also reported in a case-control study from Sweden [43,45].

The results of this study also showed no significant association between marital status and the risk of osteoporosis. A previous study conducted among 278 Korean women found no significant association between marital status and a reduction in the risk of fracture [46]. In addition, findings from a prospective study conducted in the United States with a total sample size of 5630 revealed no significant difference in the prevalence of bone fractures between married and unmarried women; however, the same study reported that the risk of fracture among married women was significantly lower than that among those who were widowed [44]. In contrast, the results from a prospective study conducted among older Mexicans showed that for Americans aged >65 years, the risk of fracture was significantly lower among married women than among unmarried women [47]. However, no significant difference in risk reduction was reported among married women compared with that among those who were divorced [47]. 

In this study, the association between osteoporosis prevalence and nutritional factors was also evaluated. Women who reported consuming calcium supplements previously or never were at a higher risk of developing osteoporosis than those who were current users. Calcium is a key element in the preservation of bone health, and lower calcium levels are expected to contribute to a higher risk of osteoporosis [20]. The NIH recommends a daily dose of 1500 mg of calcium supplements for postmenopausal women [29]. Our results support the findings of a previous study indicating that a significant increase in bone mineral density was reported among healthy postmenopausal women who were regular users of calcium supplements [48]. 

In this study, we found that the prevalence of osteoporosis was significantly higher in postmenopausal women with vitamin D deficiency than those without vitamin D deficiency. This is consistent with the findings of a previous study from Spain, where the risk of osteoporosis was increased up to five times among postmenopausal women with vitamin D deficiency [49]. Another study from Korea also reported a higher prevalence of vitamin D deficiency among postmenopausal women with fractures [50].

### Limitations

A major limitation to this study was the lack of a quantitative method for osteoporosis diagnosis, since this was a retrospective study based on previously collected data from a major national survey. In the national survey, the diagnosis of osteoporosis was based on the patient’s self-report. Another problem is the recall bias that was associated with asking participants about their dietary habits. They may not remember the number of meals they ate or the number of times they consumed vegetables or fruits during the week before the study interview. Further, unintended bias could have occurred in the selection of unemployed women, who were available at the time of the interview. On the other hand, many employed women could not participate in the study because the interviews were conducted during the day. 

## 5. Conclusions

Although the prevalence of osteoporosis in this study was different from the findings of other studies in Jordan, it highlights the need to address this issue more seriously. However, the difference in prevalence rates is more likely attributed to the different sampling methods and recognition of risk factors. Therefore, new strategies should be implemented to reduce the risk of osteoporosis among Jordanian older women. This involves creating and developing awareness programs about osteoporosis risk factors and prevention measures. Finally, additional studies and evaluations with larger samples are needed in the future. 

### Practical Implications

Managing osteoporosis can be challenging. However, reducing the risk factors and improving the lifestyle of patients is a simple and practical way to reduce the adverse effects and negative consequences of the disease, like fractures, which could lead to early disability and mortality.

## Figures and Tables

**Table 1 ijerph-19-08803-t001:** Sociodemographic characteristics of the participants.

Variable	Number of Participants	(%)
**Region** Northern Jordan Middle Jordan Southern Jordan	278 389 217	(31.4%) (44.0%) (24.5%)
**Age, years** 50–59 60–69 ≥70	649 193 42	(73.4%) (21.8%) (4.8%)
**Educational level** Primary school Secondary school College degree or higher	393 307 164	(44.5%) (34.7%) (18.6%)
**Marital status** Married Unmarried	617 265	(69.8%) (30.0%)
**Occupation** Unemployed Retired Employed	681 113 68	(77.0%) (12.8%) (7.7%)

**Table 2 ijerph-19-08803-t002:** Prevalence of osteoporosis according to demographic variables.

Variables	Osteoporosis		*p*-Value
	Present N (%)	Absent N (%)	
**Region** Northern Jordan Middle Jordan Southern Jordan	68 (24.5%) 76 (19.5%) 31 (14.3%)	210 (75.5%) 313 (80.5%) 186 (85.7%)	0.019
**Age, years** 50–59 60–69 ≥70	96 (14.8%) 63 (32.6%) 16 (38.1%)	553 (85.2%) 130 (67.4%) 26 (61.9%)	˂0.001
**Educational level** Primary school Secondary school College degree or higher	97 (24.7%) 57 (18.6%) 19 (11.6%)	296 (75.3%) 250 (81.4%) 145 (88.4%)	0.001
**Marital status** Married Unmarried	113 (18.3%) 62 (23.4%)	504 (81.7%) 203 (76.6%)	0.083
**Occupation** Unemployed Retired Employed	151 (22.2%) 15 (13.3%) 5 (7.4%)	530 (77.8%) 98 (86.7%) 63 (92.6%)	0.002

**Table 3 ijerph-19-08803-t003:** Association between osteoporosis and demographic characteristics.

Variables	Osteoporosis Present			
	OR	*p*-Value	95% CI	
			Lower	Upper
**Region** Northern Jordan Middle Jordan Southern Jordan	1 1.943 1.457	0.005 0.105	1.216 0.924	3.103 2.297
**Age, years** 50–59 60–69 ≥70	0.363 0.903 1	0.005 0.784	0.179 0.435	0.737 1.874
**Education level** Primary school Secondary school College degree or higher	1.619 1.440 1	0.157 0.287	0.831 0.736	3.155 2.818
**Occupation** Unemployed Retired Employed	2.175 1.710 1	0.123 0.330	0.811 0.581	5.838 5.031

CI, confidence interval; OR, odds ratio.

**Table 4 ijerph-19-08803-t004:** Prevalence of osteoporosis according to nutrition-related factors.

Variables	Osteoporosis		Total	*p*-Value
	Present N (%)	Absent N (%)		
Vitamin D supplement intake During the study period One week before the study period Never	41 (31.3%) 50 (22.2%) 81 (15.5%)	90 (68.7%) 175 (77.8%) 441 (84.5%)	100% 100% 100%	˂0.001
Calcium supplement intake During the study period One week before the study period Never	42 (56.8%) 64 (31.2%) 64 (10.7%)	32 (43.2%) 141 (68.8%) 533 (89.3%)	100% 100% 100%	˂0.001
Vitamin D deficiency No Yes	100 (16.6%) 72 (26.6%)	502 (83.4%) 199 (73.4%)	100% 100%	0.001
Total number of meals in the week before the study interview 1–10 11–21	165 (20.1%) 7 (13.7%)	656 (79.9%) 44 (86.3%)	100% 100%	0.267
Number of times the participants consumed vegetables/fruits during the week before the study interview 0–5 6–11	122 (19.3%) 43 (20.7%)	510 (80.7%) 165 (79.3%)	100% 100%	0.666

**Table 5 ijerph-19-08803-t005:** Association between osteoporosis and nutrition-related factors.

Variables	Osteoporosis Present			
	OR	*p*-Value	95% CI	
			Lower	Upper
Vitamin D supplement intake During the study period One week before the study period Never	1 2.161 1.388	0.008 0.180	1.227 0.860	3.807 2.239
Calcium supplement intake During the study period One week before the study period Never	1 10.272 3.823	˂0.001 ˂0.001	5.791 2.501	18.219 5.844
Vitamin D deficiency No Yes	1 1.240	0.372	0.773	1.987

CI, confidence interval; OR, odds ratio.

## Data Availability

The datasets generated and/or analyzed during the current study are available from the corresponding author on reasonable request.

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
