# Peer review of "Osteoporosis among Postmenopausal Women in Jordan: A National Cross-Sectional Study"

_ijerph, 2022, doi:10.3390/ijerph19148803_

Round 1

Reviewer 1 Report

This is a well-prepared and interesting manuscript on a significant public health problem.

Please consider some minor changes to clarify the manuscript:

1. Please provide the Ethical Board consent number (the Authors mentioned that the approval was obtained)

2. Sampling method - the Authors cited their previous paper (doi:10.1155/2018/6298739) which provides comprehensive information on sampling. Nevertheless, in this study, the Authors can add 2-3 new sentences to provide basic information on sampling and the assumptions used for sampling.

3. Please provide basic info on a questionnaire (number of items; data collection: paper-based; face-to-face interview; web-based interview). 2-3 sentences will be sufficient 

4. Please provide the limitations of this study

5. Please add 2-3 sentences on the practical implications of this study.

The results are well-presented and easy to follow. Congratulations.

Reviewer 2 Report

This is an interesting study with a good sample of women (884). Authors have made a good analysis but i have some dougbts:
1) Why have not autors questioned about the diagnostic of osteoporosis refered by participants. Based in what have women refered that they have osteoporosis. Diagnosis should be made inquantitative methods and a simple question about "have or not have" a diagnosis of osteoporosis could be an enormous bias. And with that bias results are not valid. Not reliable.

This is my main preoccupation and disconfort. 

In methodology (line 106-107) authors do not refer/explain  what are the procedures required for the study. And that should be in fact explained in metodology . Have authors performed a pre-test to the questionaire built for this study? That is imperative!

I believe also that presentation of results could be improved. In line 161 authors say that "most of the participants were unmarried". I believe that is the oposite since only 30% were unmarried and 69,8% were married. Also the reference to the respective table should be present in the text. For example in line 145-186 there are no one refrence to the tables whose data are in presentation in the text. 

One factor that is in my oppinion porly explained is the relationship between osteoporosis and unemployed/employed. Unemployed women maybe have more time to walk outdoor and take sun or do some exercise than working womens who maybe do not go out of the offices all day. Something to better discuss.

Also i believe that a good discussion should be made around the relationship between osteoporosis and educational level. There should be important reasons for those with lower educational level present higher incidence of osteoporosis. For example lower knowledge, lower food quality, poor lifestyles, etc etc . Please discuss all that. 

Round 2

Reviewer 1 Report

The manuscript was revised in line with the comments from the Reviewer.

Reviewer 2 Report

My major concern and disconfort has not changed. The way how researchers have the "diagnostic" of oteoporosis or vitamin D deficiency is a strong bias.

And participants with low level of education? How have they understand correctly what is the meaning of osteoporosis diagnosis or vit. D deficiency??  
